# Repeating fast radio burst 20201124A originates from a magnetar/Be star binary

F. Y. Wang [1,2✉], G. Q. Zhang[1], Z. G. Dai[1,3] & K. S. Cheng[4]

Fast radio bursts (FRBs) are cosmic sources emitting millisecond-duration radio bursts. Although several hundreds FRBs have been discovered, their physical nature and central engine remain unclear. The variations of Faraday rotation measure and dispersion measure, due to local environment, are crucial clues to understanding their physical nature. The recent observations on the rotation measure of FRB 20201124A show a significant variation on a day time scale. Intriguingly, the oscillation of rotation measure supports that the local contribution can change sign, which indicates the magnetic field reversal along the line of sight. Here we present a physical model that explains observed characteristics of FRB 20201124A and proposes that repeating signal comes from a binary system containing a magnetar and a Be star with a decretion disk. When the magnetar approaches the periastron, the propagation of radio waves through the disk of the Be star naturally leads to the observed varying rotation measure, depolarization, large scattering timescale, and Faraday conversion. This study will prompt to search for FRB signals from Be/X-ray binaries.

[1] School of Astronomy and Space Science, Nanjing University, Nanjing 210093, China. [2] Key Laboratory of Modern Astronomy and Astrophysics (Nanjing University), Ministry of Education, Nanjing, China. [3] Department of Astronomy, University of Science and Technology of China, Hefei 230026, China. [4] Department of Physics, The University of Hong Kong, Pokfulam Road, Hong Kong, China. ✉email: fayinwang@nju.edu.cn

The variations of Faraday rotation measure (RM)[1], an integral of the line-of-sight component of magnetic field strength and electron density, and dispersion measure (DM)[1], defined as the line-of-sight integral over the free electron density, of repeating FRBs contain essential information about their physical nature and central engine, which are as yet unknown[1–4]. Recently, the rotation measure of FRB 20201124A shows dramatically oscillations[5]. FRB 20201124A was detected by the Canadian Hydrogen Intensity Mapping Experiment (CHIME)[6] on 24 November 2020[7]. It has a dispersion measure of 413.52 pc cm$^{-3}$, larger than the Galactic DM contribution. This FRB is extremely active, with burst rate up to 50 per hour[5]. Thanks to its repetition property, it was localized to a galaxy named SDSS J050803.48 + 260338.0 at $z = 0.098$ by Australian Square Kilometre Array Pathfinder (ASKAP)[8], the Five-hundred-meter Aperture Spherical radio Telescope (FAST)[5], Very Large Array[9], and European Very Long Baseline Interferometry Network[10].

FAST detected 1863 independent bursts in a total of 88 hours from 1 April to 11 June 2021, covering 1.0 GHz to 1.5 GHz[5]. FAST observations discovered three previously unseen characteristics of repeating FRBs. First, this FRB shows dramatic Faraday rotation measure variations on a day time scale. The maximum variation of RM is about 500 rad m$^{-2}$. A small variation of RM (about 1.8%) was also found from 20 bursts observed by the Effelsberg Radio Telescope[11]. Second, it is the first repeating FRB shows circular polarization. Third, it is the repeating FRB that has the widest mean burst width[12]. These characteristics cannot be explained by current theoretical models. FRB 20201124A can be analogy to the pulsed emissions from the Galactic binary system PSR B1259-63/LS 2883[13], containing PSR B1259-63 and the Be star LS 2883. The pulsed emission of PSR B1259-63 also shows variations in its RM, DM, scattering timescale and polarized intensity on short time-scales when it approaches periastron. It has been argued that a highly magnetized neutron star in an interacting high-mass X-ray binary system can account for all of the observational phenomena of FRB 20180916B[14]. A Be/X-ray binary system, which is composed of a neutron star and a Be star with a decretion disk can well explain the frequency-dependent active window of FRB 20180916B[15].

In this work, we propose that FRB 20201124A is produced by a magnetar orbiting around a Be star companion with a decretion disk. The interaction between radio bursts and the disk of the Be star can naturally explain the varying RM, depolarization, large scattering timescale, and Faraday conversion of FRB 20201124A.

## Results

**PSR B1259-63/LS 2883**. PSR B1259-63/LS 2883 is a binary system hosting a radio pulsar and a high-mass Be star LS2883. The spin period of the pulsar is 47.76 ms and the spin-down luminosity is $8 \times 10^{35}$ erg/s[16]. Its distance is 2.6 kpc[17]. The orbital period is 1237 days with eccentricity $e = 0.9$[16–18]. LS2883 is an early Be type star that possesses a powerful stellar wind and an equatorial decretion disk[19]. There is some evidences that the Be-star disk is highly inclined to the orbital plane both from the pulsed emission[20], the unpulsed emission[21], and timing measurements[22].

This system shows unpulsed emissions in multiple bands, such as in radio[16], X-rays[23], GeV[24,25], and TeV $\gamma$-rays[26]. The light curves show double-peak profiles, which can be attributed to the interaction between the pulsar radiation and the disk[27].

Four pulsed radio observations near periastron have been made in the year of 1994[13], 1997[20], 2000[28] and 2004[29]. When the pulsar approaches the disk, the pulsed emissions of PSR B1259-63 show large variations in flux, degree of linear polarization, DM and RM[13,28,29]. These variations can be explained as the pulsed emissions interacting with the disk[13,30]. The properties of pulsed emissions, such as degree of polarization, DM and RM, are similar for the four periastron passages[29].

**Rotation measure variation**. We attempt to explain the observed properties of FRB 20201124A using a physical model for the decretion disk of a Be star (see methods, subsection The disk model). Our model is shown in Fig. 1. The Be star locates at the disk center. For the coordinate system, we set the Be star as the origin and fix the $x$-axis and $y$-axis on the disk. The orbit has an inclination $\phi$ with respect to the disk. Generally, a magnetar is formed by a supernova explosion. A supernova explosion is usually asymmetric and gives a large kick to the newly formed magnetar[31]. Due to the natal kick of the magnetar, the orbital plane and the disk plane are misaligned[32]. From observations, the inclination angle has a wide range in Be star/X-ray binary systems, from 25° to 70°[33]. In our model, after the companion dies as a supernova, a magnetar is formed, leading to FRBs generated in the magnetar magnetosphere[3,34], which is supported by the FAST observations[5]. As the orbital motion of the binary, radio bursts from the magnetar pass through different components of the disk to reach the observer, which causes the variation of RM. When the pathway of radio bursts does not intersect the disk, however, a non-evolving RM is expected.

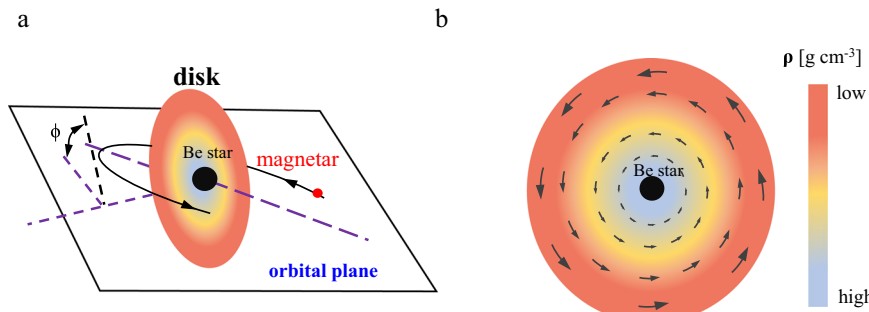

**Fig. 1 A schematic diagram of the magnetar/Be star binary model. a** The Be star locates at the center of the disk. The magnetar is shown as the red point. The stellar disk plane (shown as the black dash line) is inclined to the orbital plane (shown as purple dash lines) with the angle $\phi = \pi/12$. The purple dash line is the major axis of the magnetar's orbit. When the pathways of radio bursts pass through the disk, the interaction between bursts and disk can reproduce the observed variable rotation measure, depolarization, large scattering timescale, and Faraday conversion. **b** The face-on view of the Be star's disk. The magnetic field shown as arrows is assumed as azimuthal (or toroidal) in the disk. This model predicts that the RM contribution from the disk changes sign when the magnetar passes in front of the Be star. The RM variation discovered by FAST supports this scenario.

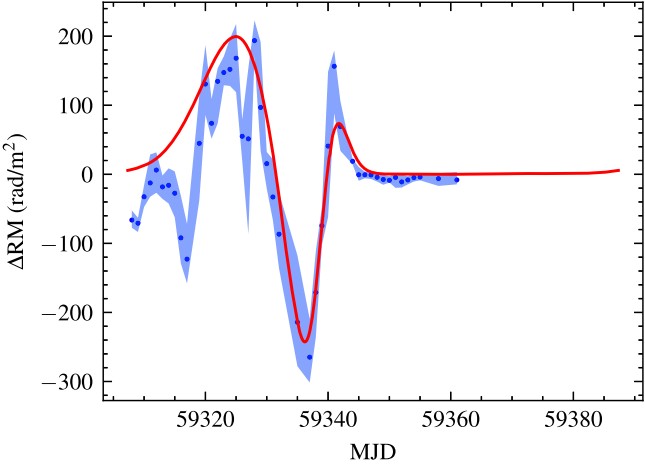

**Fig. 2 The fit of RM evolution of FRB 20201124A using the binary model.**
The blue scatter is the mean of observed $\Delta$RM in day, and the red line is the predicted evolution of our model. The blue shadow region is the range of observed RM. In the calculation, the disk density is $\rho_0 = 3 \times 10^{-14}$ g cm$^{-3}$ close to the star and has a steep-decay index $\beta = 4$. The orbital period is taken to be 80 days with orbital eccentricity $e = 0.75$. Our model can reproduce the main structure of RM evolution. However, there are some minor structures can not be reproduced. These minor structures may be caused by the clumps in the disk. Source data are provided as a Source Data file.

For the binary system PSR B1259-63/LS 2883, significant RM changes were observed both in magnitude and in sign before and after the 2004 periastron passage[29]. Compared to the Galactic PSR B1259-63, the RM variation of FRB 20201124A is small. A thin disk is considered. We assume the density of disk at the stellar surface $\rho_0 = 3 \times 10^{-14}$ g cm$^{-3}$ and $\beta = 4$, which are the typical value for disks of Be stars[35]. The orbital period is assumed to be 80 days with orbital eccentricity $e = 0.75$. Using the disk model in equation (1) and the magnetic field profile in equation (2), we can reproduce the overall RM variation with time. For the spherical coordinate origin at the Be star, the observer's direction is $\theta = 3.08$ rad and $\phi = 2.18$ rad. The derived RM variation is shown as the red line with magnetic field $\mathbf{B_0} = 10$ G at the stellar surface in Fig. 2. This magnetic field strength is consistent with those derived from RM variation and Faraday conversion[5]. The blue points is the mean observed RM variation $\Delta$RM = RM-RM$_c$ in a day. RM$_c$ is taken to be constant, as observed after MJD 59350. The blue region shows the range of $\Delta$RM in each day.

Some dips and peaks of $\Delta$RM are not well reproduced. The possible reason is that the disk is very clumpy both in density and/or magnetic field. From observations, a clumpy disk was required to explain the multi-wavelength observations of the Galactic binary system PSR B1259-63/LS 2883[36,37]. The clumpy disk is also supported by numerical simulations. For example, a two-armed spiral density enhancement caused by the tidal interaction was found using three-dimensional smoothed particle hydrodynamics simulation[38]. From observations, a stellar wind from a massive star is clumpy, with the filling factor of $f_c = 0.1$[39]. At present, there are no constraints on disk clumps. For simplicity, we assume that they are similar to the clumps in a stellar wind, without magnetic field clumps. The density inside the clumps is $1/f_c$ larger that of the smooth disk. Because the interspace between clumps is not empty, the clumping factor can be expressed as $f_c = \langle n_e^2 \rangle / \langle n_e \rangle^2$. The spatial distribution of clumps remains unknown. Using ad hoc clumps to fit the RM variation is less productive. So we discuss the effect of clumps on RM quantitatively. For the RM peak near MJD 59340, the

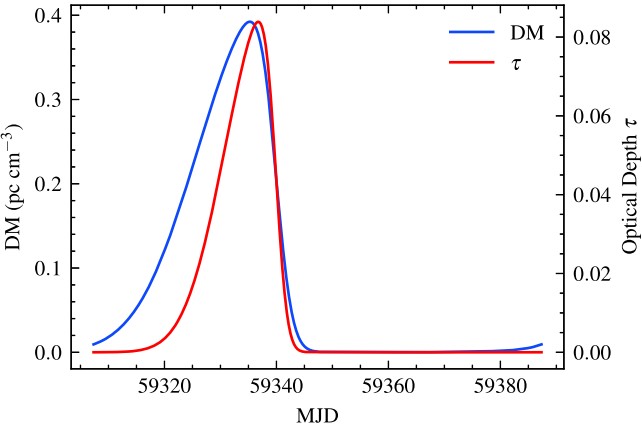

**Fig. 3 The DM and free-free absorption optical depth $\tau$ contributed by the disk.** The same parameters as Fig. 2 are used. The DM variation shown as the blue line is small, which is consistent with the FAST observation. The optical depth $\tau$ due to free-free absorption by the disk denoted as the red line is much less than 1. So the existence of the disk has no effect on the detection of radio bursts.

observed value is about ten times larger than the predication of the smooth disk. This implies the value of $f_c$ is about 0.1 if only density clumps are considered, being similar to those in a stellar wind.

The observations show that some bursts have low polarization before MJD 59350. The fluctuations of the electron density in the disk can cause differential Faraday rotation along different paths, which can account for the observed burst depolarization (see methods, subsection Depolarization in the disk). The mean burst width of FRB 20201124A is larger than the widths of other known repeating FRBs. The fluctuations of the electron density also can cause scatter broadening of the pulses (see methods, subsection Scattering timescale).

The disk also affects the DM and detection of radio bursts. In Fig. 3, we show the DM and the optical depth due to the free-free absorption contributed by the disk. The DM change shown as the blue line is small. The maximal change is 0.12 pc cm$^{-3}$. FAST found no significant DM variation in a 95%-confidence-level[5], with an upper limit $\Delta$DM $\leq 4.9$ pc cm$^{-3}$. Therefore, the model prediction is consistent with FAST observations. Next, we consider the free-free absorption as a function of orbital phase. The optical depth $\tau$ due to free-free absorption shown as the red line is less than unity. The DM and free-free absorption caused by the stellar wind can be safely ignored (see methods, subsection The effect of a stellar wind). The presence of disk clumps also affects the interaction with the radio bursts. The DM and free-free absorption are enhanced by a factor of $1/f_c = 10$ with respect to the case of a smooth disk[40]. From Fig. 3, the maximum DM contribution is 4 pc cm$^{-3}$ in the clumps, which is less than the observed DM variation[5]. After considering the disk clump, the optical depth to free-free absorption is less than 1. So the clumpy disk does not affect the detection of radio bursts. For FRBs, the induced Compton scattering should be considered[41]. The induced Compton scattering due to the disk is also estimated to be small, and does not affect FRB detection. Therefore, the disk is transparent to radio emission, and has no effect on the detection of FRBs.

**The circular polarization from Faraday conversion**. For non-repeating FRBs, variations in both linear and circular polarization have been observed[42]. However, previous observations show that repeating FRBs only have high degrees of linear polarization. FRB 20201124A is the first one showing high circular polarization. For

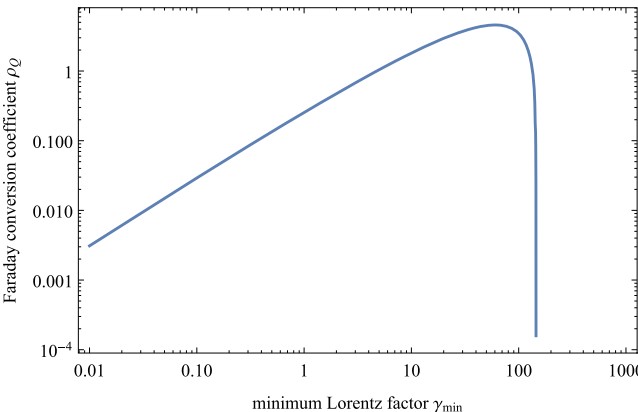

**Fig. 4 Faraday conversion coefficient $\rho_Q$ distribution as a function of minimum Lorentz factor $\gamma_{min}$.** An electron column density of $N_e = 1\,pc\,cm^{-3}$ is assumed. The plot uses $\nu = 1.25\,GHz$, $\theta = \pi/4$ and $p = 2.3$. Faraday conversion only takes effect for $\gamma_{min} < 200$.

the Galactic binary system PSR B1259-63/LS 2883, the linear and circular polarizations varied rapidly in an hour timescale near the periastron[29]. Physically, besides Faraday rotation, the propagation of FRBs through a magneto-ionic environment can also lead to Faraday conversion. In addition, a fraction of circular polarization of FRB 20201124A is produced by radiation mechanism[5]. There are two mechanisms to generate Faraday conversion. The first case is that radio signals propagate into a relativistic plasma[43]. The second case is that the magnetic field quasi-perpendicular to the wave vector is large, which had been used to interpret the circular polarization observed in solar radio bursts[44]. For the first case, termination shocks can accelerate electron and positron pairs from the magnetar wind to relativistic speeds. Because electrons and positrons contribute with the opposite sign to the circularly polarized component, the different distributions of electrons and positrons are required to generate circular polarization. In our model, a weak magnetar wind is considered. Therefore, the termination shock may be absent. For the binary system PSR B1259-63/LS 2883, the minimum Lorentz factor of the shocked electrons is estimated as[45] $10^5$. In Fig. 4, the Faraday conversion coefficient as a function of minimum Lorentz factor $\gamma_{min}$ is shown. From this figure, we can see that the Faraday conversion coefficient is found to approach zero when $\gamma_{min}$ is about 200. From the accurate calculations, the exact Faraday conversion is found to approach zero with the increasing electron energy[43], because ultra-relativistic electrons hardly interact with a radiation field. When the magnetar is in front of the Be star, FRB signals cannot interact with the plasma shocked by the termination shock, leading to no conversion. So, the Faraday conversion due to relativistic plasma is hard to account for the observed circular polarization (see methods, subsection Origin of the circular polarization).

Next we use the magnetic field quasi-perpendicular to the wave vector to explain the origin of circular polarization. When the cyclotron frequency is approaching the observation frequency, the Faraday conversion will take effect. From the FAST observations, the required magnetic field $\mathbf{B}_\perp$ is about 7 Gauss for Faraday conversion[5], which is the same order of the parallel magnetic field $\mathbf{B}_\parallel$. This mechanism naturally explains the circular polarization of FRB 20201124A. FAST observations indicate that the Faraday conversion occurs in the RM variation stage (e.g., bursts 779 and 926). When the RM variation stops, the circularly polarized bursts lack qusai-periodic structures (e.g. Burst 1472). There is no Faraday conversion. This is consistent with the prediction of our model. For the above orbital parameters, the

observed Faraday conversion appears when bursts' paths penetrate the disk. When the pathway of radio bursts does not intersect the disk, the RM variation stops. Meanwhile, there is no perpendicular magnetic field required for Faraday conversion.

**Candidates of a magnetar/Be star binary.** The total energy emitted by radio bursts during FAST observation is about $3 \times 10^{41}$ erg in 88 h[5]. For dipolar magnetic field $B_d = 10^{13}$ G, the total magnetic energy of a neutron star (NS) is $E_{mag} = 3 \times 10^{43}$ erg. In order to explain the energy budget of FRB 20201124A, about 3% NS magnetic energy is transferred to radio bursts in 88 h, which is impossible. Moreover, the Galactic FRB 20200428 suggests FRBs are accompanied by X-ray bursts (XRBs). The energy ratio between a radio burst and its associated XRB is about $10^{-5}$ [46]. If this value is valid for FRB 20201124A, the energy of XRBs is $E_X = 3 \times 10^{41}/10^{-5} \sim 10^{46}$ erg. For $E_{mag} > E_X$, the magnetic field $B_d$ is larger than $10^{15}$ G. So it should be a magnetar. The bursting activity may be related to the crustal magnetic energy of young magnetars[47]. If the converted magnetic energy comes from the toroidal component, it is at least two orders of magnitude larger than the dipolar magnetic field of the neutron star. From numerical simulations[48] and observations[49], the toroidal component could be ten times larger than the dipolar component. So, if the required magnetic field is toroidal ($>10^{15}$ G), the dipolar magnetic field is larger than $10^{14}$ G, which also supports a magnetar. Therefore, if the central engine is a compact stellar object, it should be a magentar from energy budget.

There are two candidates of a magnetar/Be star binary. The first one is LS I + 61° 303[50]. The second questionably one is SGR 0755-2933, which is also referred to as 2SXPS J075542.5-293353[51]. LS I + 61° 303 is a Be gamma-ray binary, with period about 26.496 d and a high eccentric orbit ($e = 0.537$)[52]. The mass of the Be star in this binary is about 13 $M_\odot$. It has been found that some magnetar-like bursts are possibly from this binary[50,53]. More recently, transient radio pulsed emissions from its direction were detected by FAST[54], with a period of 269 ms. This supports the presence of a rapidly rotating neutron star in this binary[54]. So the magnetar/Be binary used in this paper is close to LS I + 61° 303. They have similar orbital period and eccentricity. Searching for FRBs from LS I + 61° 303 is attractive. From the FAST observation, it has been found a rapid change of the plasma properties along the line of sight, which may be caused by the variation of the stellar wind properties[54]. Because the brightness temperatures of FRBs are much higher than that of pulsations of neutron star[55], the induced Compton scattering may be significant[41,56].

However, no FRB-like signal from the two systems has been reported. The possible reasons are as follows. First, FRBs are rare phenomena. They may be highly beamed[57,58], and have narrow spectra, making them hard to discover. For example, more than 20 magnetars have been discovered in our Galaxy. Only FRB 200428 emitted from SGR 1935+2154 has been discovered[59,60]. A bright X-ray burst is associated with FRB 200428, which has a strikingly different spectrum from other X-ray bursts[61,62]. FAST observed SGR 1935+2154 for eight hours in its active phase[57]. 29 X-ray bursts are detected, but no FRBs coincident with the bursts. Second, the high brightness temperatures of FRBs require that the emission process mechanism must be coherent[1,3,4]. The physical conditions required to produce coherent radiation in magnetars may be hard to satisfy[57]. Third, the burst rate of FRBs deviates from Poissonian and varies significantly[63-65]. For FRB 20201124A, FAST found a sudden quenching of burst activity[5]. If similar active properties of FRBs are produced in magnetar/Be star binaries, enough and proper-time observations should be performed to search FRB signals from them.

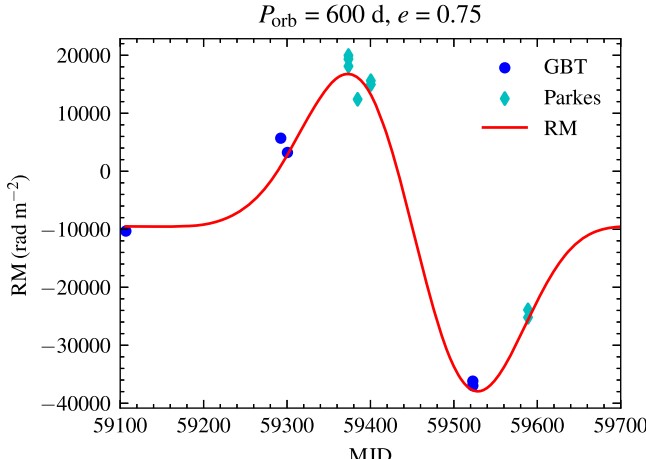

**Fig. 5 The fit of RM evolution for FRB 20190520B using the binary model.** The blue points are source RM calculated from $RM_{ob} \times (1 + z_{ob})^2$, where $RM_{ob}$ is the observed value and $z_{ob} = 0.241$ is the redshift. The red line is the fit from the binary model. The disk density is $\rho_0 = 8 \times 10^{-13}$ g/cm$^3$ with a steep-decay index of $-3$. The mass of the Be star is assumed to be $30 M_\odot$ with radius $R_\star = 10 R_\odot$. The magnetic field at the stellar surface is 100 G. The orbital period is taken as 600 d with orbital eccentricity $e = 0.75$. Source data are provided as a Source Data file.

## Discussion

Considering the interaction between a Be star disk and FRBs, we can naturally explain the mysterious features of FRB 20201124A. This model can well produce the variation of RM with a magnetic field of 10 Gauss at the stellar surface. This magnetic field and the strong electron density fluctuations can generate the observed depolarization of the FRBs by differential Faraday rotation along different paths. The magnetic field perpendicular to the wave vector in the disk, having the same order of the magnetic field along the line of sight, can account for the observed circular polarization.

Recent observations show that the RM of FRB 20190520B is rapidly varying with two sign changes[66,67], with similar magnitude as that of PSR B1259-63/LS 2883. The sinusoidal-like form of RM variation emerges. A dense magnetized screen near the FRB source can account for the RM variation and depolarization[66]. In our model, the distance of the screen can be of order the separation of the two stars. For orbital period $P = 600$ d, our model can well explain the RM variation of FRB 20190520B, which is shown in Fig. 5. This figure gives the total RM in the source frame. The RM contributed besides the disk is assumed to be $RM_{ob}(1 + z_{ob})^2 = -10301.92$ rad m$^{-2}$, where $RM_{ob} = -6700$ rad m$^{-2}$ is the first observed RM by Green Bank Telescope (GBT) at MJD 59106[66] and $z_{ob} = 0.241$ is the redshift of FRB 20190520B. The semimajor axis can be derived as $a = (GM_T P^2/2\pi)^{1/3} = 8.5$ AU, where $M_T = 30 M_\odot$ is the total mass of the binary. Our model predicts the RM evolution would be quasi-periodic. If a large number of RM detections spanning a long timescale are accumulated, the orbital period could be derived from these RM data.

## Methods

**The disk model.** There are some evidences for a disk around a Be star[68] and especially for LS 2883[19]. The disk opening angle is found to be less than 20°[69]. The disk size ranges from a few stellar radius to a few tens stellar radius from interferometry observations[35]. The density of disk can be well described by a radial power-law form, with surface density between about $10^{-13}$ to a few times $10^{-10}$ g cm$^{-3}$ [35]. From the DM and RM evolution of PSR B1259-63/LS 2883, $\rho_0 = 10^{-16}$ g cm$^{-3}$ was used[30].

The classical formation channel of a Be/X-ray binary (or a magnetar/Be binary) is as follows. In an interacting binary, the B type star accretes matter from

the companion star. Therefore, it is spun up to nearly critical velocity, leading to the formation of a decretion disk, which is supported by observations[70]. At this stage, the orbit and disk are almost in the same plane[71]. In a magnetar/Be star binary, the companion is a magnetar, which is usually formed by a supernova explosion. The process of supernova explosions is asymmetric and gives a very large kick to the newly formed magnetar[31,72]. Due to the kick of the magnetar, the orbital plane and the disk is misaligned. Besides, the kick also makes the orbit eccentric[33,73].

From observations, some Be/X-ray binaries have been found to be high misaligned[33]. For the Galactic binary PSR B1259-63/LS 2883, the inclination angle between the orbital plane and the disk plane are measured using several methods from observations. The inclination angle is found to be 10° to 40° by fitting to the observed variations in DM and RM of PSR B1259-63[30]. Using the X-ray data of PSR B1259-63, the derived inclination angel is about 70°[74]. The inclination angle of PSR B1259-63/LS 2883 system is about 25° from the optical spectra of LS 2883[75]. Using the 23 yr of pulsar timing data, the inclination angel is found to be 35°[18].

We consider the rotationally supported, geometrically thin disk model with isothermal hydrogen gas. The density profile of disk can be described as[35]

$$\rho(r, z) = \rho_0 \left(\frac{r}{R_\star}\right)^{-\beta} \exp[-z^2/H^2(r)], \qquad (1)$$

where $r$ is the distance from the star, $\rho_0$ is the disk density at the stellar radius $R_\star$, $\beta$ is the density slope, $z$ is the height, and $H(r)$ is the scale height of the disk. Observational properties of Be stars are well described by this disk model and the density slope $\beta$ is from 2 to 4[35].

The magnetic field, **B**, in the disk is poorly known. We assume that it is azimuthal and has a form

$$\mathbf{B}(r) = \mathbf{B_0}(R_\star/r), \qquad (2)$$

with $\mathbf{B_0}$ is the magnetic field strength at the stellar surface. This model can explain the RM variation of PSR B1259-63/LS 2883[13]. Given the above model, we can calculate the RM as a function of orbital phase. The RM contributed by the free electrons in the disk is

$$RM = 8.1 \times 10^5 \int_{LOS} n_e \mathbf{B_\parallel} \, dl \, \text{rad m}^{-2}, \qquad (3)$$

where $n_e$ is the free electron density in unit of cm$^{-3}$, $\mathbf{B_\parallel}$ is the magnetic field along the line of sight in unit of Gauss, and $l$ is in unit of parsec. For the azimuthal magnetic field described above, we expect the RM contributed by the disk to change sign when the magnetar passes in front of the Be star, which is supported by FAST data.

**The effect of tidal forces on the disk.** The tidal force of the magnetar could affect the disk[76]. The process of disk truncation in Be/X-ray binaries was first discussed in 1997 from the orbital period-Hα equivalent width diagram[77]. From observations, 26 Be stars show disk truncation signature[78]. Among them, six Be stars are known to have close binary companions in circular orbits[78]. In the classical viscous disk model[79], the criterion for disk truncation can be written as

$$T_{vis} + T_{res} \leq 0, \qquad (4)$$

where $T_{vis}$ and $T_{res}$ are the viscous torque and the resonant torque, respectively. Using the standard formulae of $T_{vis}$ and $T_{res}$[80], $T_{vis} + T_{res} \leq 0$ is met for a small value of $\alpha$ (viscosity parameter). The drift time-scale is $\tau_{drift} = \Delta r/v_r$, where $v_r$ is the radial velocity, and $\Delta r$ is the gap size between the truncation radius and the radius where the gravity by the magnetar begins to dominate. The spread of the disk can make the truncation inefficient for a system with $\tau_{drift}/P_{orb} = \Delta r/0.1 c_s P_{orb} < 1$. For a typical value $\alpha = 0.1$, $\tau_{drift}/P_{orb} = 0.1$ in our model, which supports the truncation effect is inefficient. This conclusion was also confirmed by numerical simulations. For misaligned Be/X-ray binary systems, smoothed particle hydrodynamics simulations show that Be star disks are not truncated in highly eccentric binaries[38]. In addition, the disk may be warped by the tidal force[38], leading to non-smooth distributions of density and magnetic field in the disk. This is the possible reason to cause the clumps in the disk. The spread of disk could cause the smaller density of the disk's outer part, compared to last periastron. Therefore, the variation of RM will become smaller in magnitude over time. In our model, the Be/magnetar binary has a highly misaligned disk, eccentric orbit and ignorable disk truncation. These three requirements are self-consistent.

Besides the tidal force, the magnetar wind may also affect the disk. In our model, the radio bursts can be powered by the magnetic-energy release of the magnetar. The magnetar wind may be weak, similar to that of Galactic magnetars (i.e., SGR 1935+2154). Strong magnetar winds can produce a bright, compact persistent radio source associated with FRBs[81,82], which has been observed in FRB 20121102A[83] and FRB 20190520B[84]. However, there is no compact persistent radio emission associated FRB 20201124A[10]. The extended radio emission around FRB 20201124A is attributed to star-formation activity[8,9,85].

**Fitting RM using the binary model.** Compared to the Galactic PSR B1259-63, the RM variation of FRB 20201124A is small. We consider a thin disk by assuming $\rho_0 = 3 \times 10^{-14}$ g cm$^{-3}$, and $\beta = 4$. The radius and mass of Be star is taken to be

$R_\star = 5R_\odot$ and $M_\star = 8M_\odot$. Typical velocities of the plasma in the disk are[86] $v_d = 150-300$ km s$^{-1}$. The vertical scale height is derived from

$$H(r) = c_s \left( \frac{r}{GM_\star} \right)^{1/2} r, \quad (5)$$

where $c_s$ is the sound speed. For an isothermal gas, the sound speed is $c_s = (kT/\mu m_H)^{1/2}$, where $T$ is the isothermal temperature, $\mu$ is the mean molecular weight, and $m_H$ is the mass of hydrogen. We assume $\mu = 1$ and $T = 10^4$ K. In order to prevent that the disk is significantly truncated by tidal forces, we consider a large eccentric orbit with $e = 0.75$. In our model, the RM variation is mainly affected by the orbital motion. We assume the orbit period is $P = 80$ days. Then the semi-major axis can be derived from

$$a_0 = \left[ \frac{G(M_\star + M_{NS})P^2}{4\pi^2} \right]^{1/3}, \quad (6)$$

where $M_{NS} = 1.4M_\odot$ is the magnetar mass. To match the observed RM evolution, we assume that the inclination angle is $\phi = \pi/12$ and the observer's direction is $\theta = 3.08$ rad and $\phi = 2.18$ rad. We assume the magnetic field $B_0 = 10$ G. From observations, the surface magnetic field of a Be star has been measured. The magnetic field in the longitudinal direction is between 2 and 300 Gauss for 15 Be stars[87]. From a large sample of 60 Be stars, the maximal surface magnetic field 4000 Gauss was derived[88]. Therefore, the value of $B_0 = 10$ G is well in the allowed range.

The DM contributed by the disk is

$$DM = \int_{LOS} n_e dl, \quad (7)$$

where $n_e$ is the free electron density along the line of sight. In the observations, FAST can detect FRB signals. Therefore, the optical depth

$$\tau = 8.2 \times 10^{-2} T^{-1.35} \nu^{-2.1} \int_0^d n_e^2 dl \quad (8)$$

of free-free absorption must be less than unity. In the above equation, $T$ is the temperature in Kelvin, $\nu$ is the observing frequency in GHz, and $d$ is the distance to the magnetar (in parsec). The same parameters for the disk is used to calculate the DM and optical depth.

The induced Compton scattering is crucial for high brightness temperature sources. In our model, the scattering occurs in the nearest vicinity of the source. For a single narrow beam of uniform brightness and solid angle $\Delta\Omega$, the effective optical depth is[41,56]

$$\tau_{ind} = \frac{kT_B}{m_e c^2} (\Delta\Omega^2) \tau_T, \quad (9)$$

where $T_B$ is the brightness temperature, and $\tau_T = n_e(r)\sigma_T r$ is the Thompson scattering optical depth. For radio bursts with duration $w$ and flux $S$ occurring at redshift $z = 0.098$, the brightness temperature is

$$T_B = Sd^2/2k(\nu w)^2 \sim 6.6 \times 10^{30} \text{ K} \left( \frac{S}{Jy} \right) \left( \frac{\nu}{GHz} \right)^{-2} \left( \frac{w}{ms} \right)^{-2} \left( \frac{d}{450Mpc} \right)^2. \quad (10)$$

The radiation subtends a solid angle $\Delta\Omega = (cw)^2/r_d^2 = 10^{-10}$ with the distance $r_d = 50R_\odot$ between the magnetar and the disk. If we assume radio bursts travel at distance $2R_\star$ away from the Be star, using the base density $5 \times 10^{-14}$ g cm$^{-3}$ of the disk and equation (9), we can estimate $\tau_{ind}$ is 0.025, which is much less than one. Therefore, radio bursts can escape from the disk.

**The effect of a stellar wind**. A massive star usually has a large wind mass loss, which strongly depends on the stellar effective temperature[89]. The effect of a stellar wind on DM and free-free absorption should be considered. For an isotropic wind, the number density of the wind is given by

$$\rho_{wind} = \rho_{0,wind}(r/R_\star)^{-2}, \quad (11)$$

where $r$ is the radial distance from the star. The stellar surface density is $\rho_{0,wind} = \dot{M}/4\pi R_\star^2 v_w$ with mass loss rate $\dot{M}$ and wind velocity $v_w = 3 \times 10^8$ cm s$^{-1}$. From observations, the mass loss rate of Be stars has been estimated to be[90,91] $10^{-11}-10^{-8}M_\odot$ yr$^{-1}$. A typical value $\rho_{0,wind} = 4.18 \times 10^{-17}$ g cm$^{-3}$ is adopted, which corresponds to the mass loss rate $3 \times 10^{-10}M_\odot$ yr$^{-1}$. The ratio of the mass flux density in the disk over the stellar wind is

$$\frac{\rho_0 v_d}{\rho_{0,wind} v_w} \sim 80, \quad (12)$$

which is well in range of 30–100[92].

Because of adiabatic cooling, the wind temperature decreases slowly with radial distance from the star. The temperature of the stellar wind follows a power-law evolution due to adiabatic cooling[93]

$$T_{wind} = T_{0,wind}(r/R_\star)^{-\beta_1}. \quad (13)$$

In our calculation, we use the effective temperature of star $T_{0,wind} = 3 \times 10^4$ K and $\beta_1 = 2/3$. In this case, the maximum $DM_w$ and $\tau_w$ contributed by the stellar wind are 0.54 pc cm$^{-3}$ and 0.005, respectively. Based on these results, the

contribution of the stellar wind can be safely ignored. We are not aware of magnetic field measurements in the wind of a massive star. For the Galactic PSR B1259-63, the RM variation cannot be explained by the contribution from the stellar wind[30]. The geometry of the magnetic field in the stellar wind may be toroidal[30,94]. Therefore, the radio burst propagates orthogonally to the toroidal magnetic field in a significant fraction of the orbit, which contradicts the observed RM evolution.

Below, we compare the magnetic field in the pulsar wind and in stellar disk. Since the magnetic field is dipolar inside the magnetar light cylinder, $r_{LC} = 5 \times 10^9 P_{m,s}$ cm, and toroidal ($\propto r^{-1}$) in the wind, the magnetic field strength at a distance $r$ is

$$B_{wind} = 230B_{14}P_s^{-3} \left( \frac{5 \times 10^9 \text{ cm}}{r} \right) \text{G}, \quad (14)$$

for the period of magnetar $P_m = 2$ s. We make the comparison at the same radius $r = 2R_\star = 10R_\odot$. $B_{wind} = 0.9$ G is less than the magnetic field in the disk. Therefore, the magnetic field in the disk dominates.

**Depolarization in the disk**. The observed pulse depolarization is attributed to random fluctuations in the magnetic field and/or electron density in the disk, which can cause differential Faraday rotation along different paths of the light rays[95]. If the Faraday dispersion function is well described by a Gaussian distribution with variance $\sigma^2 = k^2 \langle \delta(RM)^2 \rangle$, the measured polarized intensity can be parameterized as

$$P(\lambda^2) = P_i \exp[-2k^2 \langle \delta(RM)^2 \rangle \lambda^4], \quad (15)$$

where $P_i$ is the intrinsic polarized intensity, $\lambda$ is the observing wavelength, $k = 0.81$ is a constant, and $\langle \delta(RM)^2 \rangle^{1/2}$ is the variance in $\Delta RM$. For simplicity, we assume that $\langle \delta(RM)^2 \rangle^{1/2}$ arises from fluctuations in free electron density $n_e$. Furthermore, we assume that the variance of electron density $n_e$ satisfies $\langle \delta n_e^2 \rangle^{1/2} \propto n_e$ in the disk[30], which is similar to the small-scale turbulence in the interstellar medium. Letting $\langle \delta(RM)^2 \rangle^{1/2} = f\Delta RM$, the stochastic Faraday rotation leads to the depolarization when $|\Delta RM| > f^{-1}\lambda^{-2}$. From the FAST observation, the value of $|\Delta RM|$ is about 200 rad m$^{-2}$ at frequency 1.25 GHz. Therefore, $f$ is about 0.1. We conclude that stochastic Faraday rotation may be responsible for the observed depolarization.

**Scattering timescale**. From the CHIME observation, the largest scattering time-scale of FRB 20201124A is 14 ms[7]. The fluctuations of the electron density in the disk can cause scatter broadening of the pulses. The electron plasma with density fluctuations of $\delta n_e$ and spatial scale $a$ produces a typical scattering angle at wavelength $\lambda$ of

$$\theta_s = \frac{1}{2\pi} \left( \frac{L}{a} \right)^{1/2} r_0 \lambda^2 \delta n_e, \quad (16)$$

where $r_0$ is the classical electron radius and thickness of the screen $L$ is assumed to be roughly equal to its distance $d$ from the magnetar[96]. If the scattering is due to a thin screen at a distance $r_s$ from the magnetar, the scattering angle $\theta_s$ is $(2\tau_s c/r_s)^{1/2}$, with the scattering time $\tau_s$. We use the burst 20210405B with scattering time $\tau_s = 12.17$ ms detected by CHIME[7] to estimate the spatial scale $a$. The distance between the burst and the screen is about 42 solar radius. Therefore, the scattering angle $\theta_s$ is about 0.9°. From equation (16), the density fluctuation of electron plasma in the disk is $\delta n_e = 80a^{1/2}$ cm$^{-3}$. If $\delta n_e$ is set to be equal to the total electron density $n_e$ in the scattering region, from the disk model, then $\delta n_e = 3.9 \times 10^6$ cm$^{-3}$ is found, implying that $a = 2.4 \times 10^9$ cm.

**Origin of circular polarization**. FRB 20201124A is the first repeating FRB with significant circular polarization as discovered by FAST[5] and Parkes[97]. Two possible mechanisms can cause Faraday conversion: linearly polarized light is converted to circularly polarized one and vice versa. The polarization measurement of FRB 121102 has been used to constrain the local magneto-ionic environment using Faraday conversion[98,99].

First, we consider that the circular polarization arises from the propagation through mildly relativistic plasma. For the PSR B1259-63/LS 2883, X-ray, GeV gamma-ray and Tev gamma-ray radiations have been observed. The natural physical mechanism for these high-energy emissions are as follows. An interaction between the relativistic wind of the pulsar and the low-velocity wind from the Be star can form a termination shock. The electron and positron pairs from the pulsar wind are accelerated to relativistic velocities by the terminal shock and emit broadband non-thermal emission through synchrotron radiation and inverse Compton scattering. In our model, a termination shock may not exist, due to the weak magnetar wind.

If only the relativistic plasma is considered, the change of polarization angle is

$$\Delta\phi = \frac{1}{2} RRM \lambda^3, \quad (17)$$

where RRM is the relativistic rotation measure (RRM), and $\lambda$ is the observed wavelength. We assume a power-law distribution for accelerated particles

$$N(\gamma) = N_0 \gamma^{-p}, \quad \gamma_{min} \leq \gamma \leq \gamma_{max}, \quad (18)$$

where $\gamma_{min}$ and $\gamma_{max}$ are the minimum and maximum Lorentz factors, respectively. For the power-law index $p \neq 1$, the normalization factor is $N_0 = n_r(p-1)(\gamma_1^{1-p} - \gamma_2^{1-p})^{-1}$ with the number density $n_r$ of the relativistic particles. The definition of RRM is[100]

$$RRM = \frac{e^4}{4\pi^3 \epsilon_0 m_e^3 c^4} \frac{p-1}{p-2} \int n_r (\mathbf{B} \sin\theta)^2 \gamma_{min} ds. \quad (19)$$

The fraction of circular polarization is $\Pi_c = RRM\lambda^3$ for the relativistic plasma dominated case. For relativistic plasma, the definition of $DM_r$ is $DM_r = DM \times \mathcal{D}$[101], with Doppler factor $\mathcal{D} = [\gamma(1 - v/c \cos\theta)]^{-1}$, where $\theta$ is the angle between the line of sight and the direction of plasma motion. The RRM effect should require $n_e \ll n_r \gamma_1 \ln(\nu/f_c \gamma_1)$.

For the 1994 periastron passage of PSR B1259-63, the changes of DM and RM cannot be explained in the termination shock model[30]. The observed changes of DM in four periastron passages of PSR B1259-63 are roughly consistent with the contributions from the disk and wind of Be star, with the disk component dominated[102]. Therefore, the cold plasma dominates the wave properties. The RRM effect is very weak. When the magnetar is in front of the Be star, radio bursts cannot interact with the plasma shocked by the termination shock. Therefore, there is no Faraday conversion, which is disfavored by the FAST observations.

Numerically, the approximated Faraday conversion coefficient is[43,103]

$$\rho_Q = 8.5 \times 10^{-3} \frac{2}{p-2} \left[ \left( \frac{\omega}{f_c \sin\theta \gamma_{min}^2} \right)^{(p-2)/2} - 1 \right] \frac{p-1}{\gamma_{min}^{1-p}} \left( \frac{f_c \sin\theta}{\omega} \right)^{\frac{p+2}{2}} \frac{2\pi}{\omega}, \quad (20)$$

where $\omega = 2\pi\nu$, $f_c = e\mathbf{B}/m_e c$ is the cyclotron frequency, and $\theta$ is the angle between wave vector and the line of sight. This approximation approaches zero if $\gamma_{min} \geq 200$. In order to explain the high-energy emission from the Galactic binary system PSR B1259-63/LS 2883, an extreme large $\gamma_{min} = 10^5$ is required[45,102]. Therefore, the relativistic plasma hardly generates the observed circular polarization, unless the minimum Lorentz factor is less than 100.

Next we consider the Faraday conversion occurring when the magnetic field is perpendicular to the wave propagation direction. The required magnetic field for Faraday conversion is about 7 G[5], which is the same order of the magnetic field strength along the line of sight. The Be star disk can provide the required vertical magnetic field. This is the most natural way to explain the production of circular polarization.

## Data availability

The datasets generated during and/or analysed during the current study are available from the corresponding author on reasonable request. Source data are provided with this paper.

## Code availability

The code used in this study is freely available for download at https://github.com/astrogqzhang/FRB_RM_Evolution_BINARY, also available at ZENODO repository https://zenodo.org/badge/latestdoi/491122370.

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

## Acknowledgements

We thank B. Zhang, Y. Shao and K.J. Lee for helpful discussions on the binary model, H. Xu for providing the RM data, and J.P. Hu for drawing Fig. 1. This work was supported

by the National Natural Science Foundation of China (grants No. U1831207 (F.Y.W.) and 11833003 (Z.G.D.)), the Fundamental Research Funds for the Central Universities (Nos. 0201-14380045 (F.Y.W.)), the China Manned Spaced Project (CMS-CSST-2021-A12 (F.Y.W.)), the National SKA Program of China (grant No. 2020SKA0120300 (Z.G.D.)), and the National Key Research and Development Program of China (grant No. 2017YFA0402600 (Z.G.D.)). This work made use of data from FAST, a Chinese national mega-science facility built and operated by the National Astronomical Observatories, Chinese Academy of Sciences.

## Author contributions

F.Y.W. initiated the study. G.Q.Z. performed the RM fit with the help of F.Y.W. Z.G.D. and K.S.C. joined the model discussion. F.Y.W. wrote the draft, with contributions from G.Q.Z., Z.G.D., and K.S.C.

## Competing interests

The authors declare no competing interests.
