## [Peer Review File · Nature Communications]

REVIEWER COMMENTS

Reviewer #2 (Remarks to the Author):

I have read the new version of the paper "A repeating fast radio burst residing in a magnetar/Be star binary", by Wang et al. I find that the authors have considered my earlier comments seriously and believe that they have produced a self-consistent model. New data on the RM variation of FRB 20190520B from January 2022 (ATel #: 15190, 15192) can be described by their model as they show in the reply. For all this, I would have no problem seeing this paper published in *Nature Communications*. It will attract further discussion. I will remark just a couple of points to help the editors reach a decision.

-I continue to think that the proposed idea might indeed work, but there is not observational proof for it yet. The authors have done a great effort in this version, clarifying the needed (simplifying) assumptions in which this model would work, and this is now more clearly showing how these assumptions are embedded in the conclusions. Suffice to look at the discussion on clumping and truncation. Not only the binary should accommodate a significantly high disk inclination, but also this disk should not be prone to truncation. Both are in principle possible for ranges of parameters or particular supernova kicks, but in my opinion the detection of FRBs cannot be taken as an a posteriori proof of such conditions.

-The authors emphasize both in the paper (from the title onwards, of course) and in the reply, that the binary contains a magnetar, useful to further differentiate this model from similar others published before. The magnetic field required in their model must be higher than $1E14G$, as they claim. But they are referring to a dipolar field, which, after the discovery of low-field magnetars is known not to be uniquely relevant. In particular, the authors consider in the reply that given the total energy emitted by radio bursts in 59 days is $3E41$ erg, and that this is $\sim 3\%$ of a $1E13$ G (dipolar) magnetic energy, such neutron star could not be part of the system. But if converted magnetic energy come from a toroidal component, maintaining the dipolar one relatively low would be possible.

Reviewer #2 (Remarks to the Author):

I have read the new version of the paper "A repeating fast radio burst residing in a magnetar/Be star binary", by Wang et al. I find that the authors have considered my earlier comments seriously and believe that they have produced a self-consistent model. New data on the RM variation of FRB 20190520B from January 2022 (ATel #: 15190, 15192) can be described by their model as they show in the reply. For all this, I would have no problem seeing this paper published in Nature Communications. It will attract further discussion. I will remark just a couple of points to help the editors reach a decision.

-I continue to think that the proposed idea might indeed work, but there is not observational proof for it yet. The authors have done a great effort in this version, clarifying the needed (simplifying) assumptions in which this model would work, and this is now more clearly showing how these assumptions are embedded in the conclusions. Suffice to look at the discussion on clumping and truncation. Not only the binary should accommodate a significantly high disk inclination, but also this disk should not be prone to truncation. Both are in principle possible for ranges of parameters or particular supernova kicks, but in my opinion the detection of FRBs cannot be taken as an a posteriori proof of such conditions.

Re: Thanks for the comments. In our model, the Be/magnetar binary has a highly misaligned disk, eccentric orbit and ignorable disk truncation. These three requirements are self-consistent. Due to the natal kick of magnetar, the orbital plane and the disk is misaligned. Meanwhile, the kick from the supernova also makes the orbit be eccentric, and even unbinds the binary (Martin, Tour & Pringle 2009, MNRAS, 397, 1563). As the reviewer said, both the misaligned disk and highly eccentric orbit are possible from supernova kicks. In the misaligned disk and eccentric orbit case, the disk truncation effect is inefficient, which was confirmed by theoretical (Okazaki & Negueruela 2001), observational (Negueruela & Okazaki 2001) and numerical studies (Okazaki et al.

2011). So, these three requirements (misaligned disk, eccentric orbit and ignorable disk truncation) are self-consistent. If the orbit is circular and the disk and orbit is coplanar, the disk will be truncated significantly, which has been confirmed by numerical simulations (Okazaki et al. 2002; Cyr et al. 2017), semi-analytical (Okazaki & Negueruela 2001) and observations (Reig et al. 1997).

From observations, some Be/X-ray binaries have been found to be high misaligned with eccentric orbit, such as PSR 0053+604 and PSR J0045–7319 (Martin et al. 2009), especially the Galactic binary PSR B1259-63/LS 2883. So, the requirements our proposed model are not special. The whole proposed idea is supported by theoretical and observational studies.

Some discussions are added in pages 19 and 20.

-The authors emphasize both in the paper (from the title onwards, of course) and in the reply, that the binary contains a magnetar, useful to further differentiate this model from similar others published before. The magnetic field required in their model must be higher than 1E14G, as they claim. But they are referring to a dipolar field, which, after the discovery of low-field magnetars is known not to be uniquely relevant. In particular, the authors consider in the reply that given the total energy emitted by radio bursts in 59 days is 3E41 erg, and that this is ~3% of a 1E13 G (dipolar) magnetic energy, such neutron star could not be part of the system. But if converted magnetic energy come from a toroidal component, maintaining the dipolar one relatively low would be possible.

Re: Thanks for the comments. The total energy emitted by radio bursts in 88 hr is

$$E_{\text{radio}} = 20 \text{ hr}^{-1} * 24 \text{ hr/day} * 59 \text{ day} * 10^{37} \text{ erg} \sim 3 * 10^{41} \text{ erg}.$$

For $B = 10^{13}$ G, the magnetic energy of NS is $E_{\text{mag}} = B^2 * R^3 / 6 \sim 3 * 10^{43}$ erg, which $R = 12$ km is the NS radius. In order to explain the energy of FRBs, about 3% NS magnetic energy is transferred to radio bursts in 88 hr, which is impossible. Moreover, the Galactic FRB 20200428 (CHIME/FRB 2020, Nature) suggests the FRBs are accompanied by X-ray bursts (XRBs). The energy ratio between radio burst and XRB

is about 10^{-5} . If this value is valid for extra-galactic FRB 20201124A, the X-ray energy release for FRB 20201124A is $E_X = 3 \times 10^{41} / 10^{-5} \sim 10^{46}$ erg. For $E_{\text{mag}} > E_X$, the magnetic field B is larger than 10^{15} G. So it should be a magnetar. If the converted magnetic energy comes from a toroidal component, the toroidal magnetic field is at least two orders of magnitude larger than dipole magnetic field. From numerical simulations (Braithwaite & Nordlund 2006) and observations (Igoshev et al. 2021), the toroidal component could be ten times larger than dipolar component. So if the required magnetic field is toroidal ($>10^{15}$ G), the dipolar one is larger than 10^{14} G, which supports a magnetar.

A paragraph is added in page 27 to clarify this point.

REVIEWERS' COMMENTS

Reviewer #2 (Remarks to the Author):

The authors have reasonably considered my suggestions and I thus recommend accepting the paper. A comment: Regarding the second point, I wanted to convey that the dipolar component of the field, on which most FRB-magnetar models are currently based, plays a minor role -if any- in determining the bursting activity (see e.g., Dehman et al. 2022, ApJ Letters 902, id.L32).

Reviewer #2 (Remarks to the Author):

The authors have reasonably considered my suggestions and I thus recommend accepting the paper. A comment: Regarding the second point, I wanted to convey that the dipolar component of the field, on which most FRB-magnetar models are currently based, plays a minor role -if any- in determining the bursting activity (see e.g., Dehman et al. 2022, ApJ Letters 902, id.L32).

Re: Thanks for the suggestion. We have added a sentence in page 28 to clarify this point. The paper of Dehman et al. (2022) is added to the reference list.